# Sustainable Strategies for Increasing Legume Consumption: Culinary and Educational Approaches

**DOI:** 10.3390/foods12112265

**Published:** 2023-06-04

**Authors:** Isaac Amoah, Angela Ascione, Fares M. S. Muthanna, Alessandra Feraco, Elisabetta Camajani, Stefania Gorini, Andrea Armani, Massimiliano Caprio, Mauro Lombardo

**Affiliations:** 1Department of Biochemistry and Biotechnology, Kwame Nkrumah University of Science and Technology, Kumasi 0023351, Ghana; isaacamoah@knust.edu.gh; 2Department of Human Sciences and Promotion of the Quality of Life, San Raffaele Open University, Via di Val Cannuta, 247, 00166 Rome, Italy; angela.ascione@uniroma5.it (A.A.); alessandra.feraco@uniroma5.it (A.F.); elisabetta.camajani@uniroma5.it (E.C.); stefania.gorini@uniroma5.it (S.G.); andrea.armani@uniorma5.it (A.A.); massimiliano.caprio@uniroma5.it (M.C.); 3Pharmacy Department, Faculty of Medicine and Health Sciences, University of Science and Technology-Aden, Alshaab Street, Enmaa City 22003, Yemen; f.mothana@aden.ust.edu; 4Laboratory of Cardiovascular Endocrinology, San Raffaele Research Institute, IRCCS San Raffaele Roma, Via di Val Cannuta, 247, 00166 Rome, Italy

**Keywords:** diet, cooking, legumes, consumption, intake, nutrient profile, health effects, vegetables, physiological phenomena, fabaceae

## Abstract

Legumes are nutrient-dense crops with health-promoting benefits. However, several barriers are associated with their consumption. Emerging issues including food neophobic tendencies or taboos, unclear dietary guidelines on legume consumption, health concerns, and socio-economic reasons, as well as long cooking procedures, adversely affect legume consumption frequency. Pre-treatment methods, including soaking, sprouting, and pulse electric field technology, are effective in reducing the alpha-oligosaccharides and other anti-nutritional factors, eventually lowering cooking time for legumes. Extrusion technology used for innovative development of legume-enriched products, including snacks, breakfast cereals and puffs, baking and pasta, represents a strategic way to promote legume consumption. Culinary skills such as legume salads, legume sprouts, stews, soups, hummus, and the development of homemade cake recipes using legume flour could represent effective ways to promote legume consumption. This review aims to highlight the nutritional and health effects associated with legume consumption, and strategies to improve their digestibility and nutritional profile. Additionally, proper educational and culinary approaches aimed to improve legumes intake are discussed.

## 1. Introduction

The world population is constantly growing, and the current global population of 8 billion (as of mid-November 2022) is expected to increase to 9.7 billion in 2050 [1]. Increased population pairs with a predicted higher demand for food. Total food demand, including that from protein sources, is expected to increase from 35% in 2010 to 56% in 2050 [2]. In the past decade, consumers’ interest in increasing protein intake from alternative sources, including plant-based types such as legume-derived ones, has increased. This request has been driven by factors including the relatively lower carbon footprint associated with leguminous crop cultivation, with its fewer deleterious effects on climate change compared to cattle breeding [3,4,5], health reasons [3,5], and the relatively high cost of meat compared to legume-derived proteins [6], especially in Southeast Asia and Sub-Saharan Africa. Moreover, the increased sustainability associated with the production of plant crops, including legume proteins, should be taken into account [7,8]. Globally, different legume types are widely distributed. 

The Food and Agriculture Organization of the United Nations (FAO) recognizes ten main types of legumes: dry beans including soybean, dry broad beans, dry peas, chickpeas, cowpeas, pigeon peas, lentils, Bambara beans, vetches, and lupins [9]. Additionally, there are “minor” legumes that are less commonly used internationally [9]. The four most commonly used species for food purposes, marketed in both dry and hydrated forms, are beans, lentils, peas, and chickpeas. Leguminous crops are resilient towards climate changes and possess an inherent ability to fix atmospheric nitrogen in the soil leading to improvement in soil fertility [10]. In terms of nutritional composition, legumes contain a high concentration of protein (23.7–25.9 g/100 g dry beans and 8.23–9.01 g/100 g for cooked beans), a high content of soluble and insoluble fibers (4.1–4.3 g/100 g dry beans and 7.0–10.5 g/100 g for cooked beans), a low lipid content (except for soybeans), are free of saturated fats and cholesterol, and are good sources of B-group vitamins, iron, magnesium, copper, selenium, calcium, potassium, zinc, and manganese [11,12]. Legumes generally have a low caloric density and low glycemic index, due to their high fiber composition [12].

According to European and American nutritional guidelines, legumes are an important part of a healthy diet [13,14]. The European Food Safety Authority [15] states that legumes represent a valid source of protein and that they are essential for a healthy diet. It has been recommended that adults consume at least 20–25 g of protein from plant sources per day [13]. Due to the excellent nutritional profile and health-promoting effect associated with legume consumption, they represent an important component of several dietary guidelines. For example, increased consumption of legumes has been encouraged as part of the Mediterranean and the DASH dietary pattern guidelines. Several epidemiological studies demonstrated improved outcomes in cardiovascular disease, diabetes, cancer, Alzheimer’s disease, and Parkinson’s disease following strict adherence to the Mediterranean and DASH diet, both including legumes [16]. The DASH diet recommends an intake of about four-to-five servings of legumes per week for a 2000-calorie diet to effectively control blood pressure [17]. Additionally, the 2020–2025 Dietary Guidelines for Americans recommend legumes as part of a healthy dietary pattern, stating that “legumes provide protein, dietary fiber, and a variety of vitamins and minerals, including iron, potassium, and folate” [14]. Legumes are a healthy and versatile food option for people following specific diets, such as those for people with gluten intolerance [18] or vegetarianism [19]. This has been largely attributed to their gluten-free nature [20]. Furthermore, legumes combined with cereals provide all of the essential amino acids and represent a valuable one-dish meal for vegans or vegetarians. Legumes are thus used in complementary formulations in cereal-based products, due to their high amino acid lysine content which in cereals is deficient [21]. 

Hughes et al. [22] recently showed that there was a general global decline in beans and legume consumption, according to data derived from the Global Dietary Database (“GDD 2018”) in June 2022. The authors evaluated the national intake of legumes and beans considering the 50 g/day [23,24] target, which has been established according to several meta-analyses as the ideal intake required to reduce mortality and morbidity outcomes [23,24]. The general decline observed could be attributed to the lack of clear recommendations on the appropriate quantity and serving size of beans and other legume-based products despite the availability of national food-based dietary guidelines, as observed in Australia [25]. This is because most of the food-based dietary guidelines group together legumes with seeds, vegetables, and starchy staples [25]. In a recent cross-national survey conducted in Europe, consumers from Poland, Spain, and Germany reported that digestion challenges associated with legume consumption were an important barrier that restricts legume intake. In Danish and UK consumers, the lack of available easy-to-cook legume and legume-based products was the underpinning barrier to their consumption [26]. Consequently, there is a need to implement strategies that could promote legume intake in the population. This could lead to improved health outcomes and increased sustainable food supply, resulting in a lower carbon footprint and leading to better environmental sustainability. In 2015, Polak et al. [27] reviewed the health effects and some culinary approaches that could be adopted to promote legume consumption. However, barriers and strategies to improve digestibility and overcome digestive challenges associated with legume intake, leading to increased consumption, have not been addressed. The present article therefore highlights the effects of cooking on the nutritional profile of legumes, underlines health effects associated with legume consumption, and, finally, examines strategies to improve the digestibility and nutritional profile of legumes. Additionally, the use of state-of-the-art educational and culinary approaches that could be exploited to improve legume intake are extensively discussed. 

### 1.1. Nutritional Composition of Commonly Consumed Legumes 

The nutritional composition of legumes influences their health-promoting properties, which is an important factor that can influence consumers’ consumption of legumes. Legumes are rich in protein and fiber. For example, according to the USDA Food Data Central database, protein and fiber content ranging from 19.0 to 36.0 g/100 g and from 9.0 to 25.0 g/100 g, respectively, for legumes including common beans, lentils, chickpeas, peas, broad beans, and soybeans have been reported [28] (Table 1). 

The high protein content of legumes is a promising trait that makes them ideal candidates for product reformulation with foods that are generally lower in protein. It is important to note that protein addition would influence aliment techno-functional and sensorial properties when used in food product development. Although plant-based proteins are deemed second-class proteins, efforts to improve their digestibility could enable them to compete with those derived from animal sources. The relatively high fiber content [28] makes them promising for lowering the glycemic impact of foods when incorporated into other staple foods and could induce satiety. The cooking of legumes is essential as it impacts its digestibility and nutritional composition [29]. Storz et al. [30], recently, using a nationally representative sample of 9078, investigated the current sex distribution of cooking responsibilities in the United States of America. The authors reported that cooking duties are still being mostly performed by women. Consequently, women could be targeted for educational training efforts to promote legume consumption. The effect of cooking methods on the nutritional composition of some commonly consumed legumes will be discussed. 

Margier et al. [31] investigated the effect of household cooking and canning on the nutrient profile of common legumes including kidney beans, white beans, chickpeas, and brown and green lentils. For the household cooking employed, the authors adopted household cooking methods protocol established by the French National Federation of Dry Legumes. The legumes were soaked in low-ionized water overnight at room temperature using a water-to-legume ratio of 5:1, with the exception of lentils. The legumes were drained of the water and cooking proceeded afterwards using a water-to-legume ratio of 2:1. Cooking duration was 25 min, 1 h 30 min, and 2 h for the lentils, kidney beans/white beans, and chickpeas, respectively. In the canning process, the pre-treatment method, which involved an overnight legume soaking in water at a ratio of 1:3, was employed. Blanching legumes at 90 °C for 5 min was carried out prior to transfer into a can with hot brine at a ratio of 195/236 (*w*/*w*). Sterilization of the legumes was carried out at 127 °C for 16 min in the can. The final step involved the cooling of the sterilized products to a temperature of 30 °C for 10 min. The authors stabilized the sterilized legumes by draining the water, rinsing the legumes in water, and subsequently freezing the legumes. The samples were freeze-dried and kept at a temperature of −80 °C until analysis. The results showed a general increase in the protein and fiber content of all the legumes cooked using the household cooking method compared to the canning method. Specifically, the protein and fiber content of kidney beans, white beans, chickpeas, and lentils increased by 30.30 and 43.97%, 17.78 and 33%, 17.44 and 21.95%, and 38.87 and 55.29%, respectively, when the household cooking method was used, compared to the sterilization process [31]. The general decrease in protein concentration of the cooked legumes compared with the raw types could be attributed to the loss of proteins into the cooking water, which could be considered part of the cooking losses.

### 1.2. Bioactive Properties of the Minor Components of Legumes 

Legumes, aside their rich nutrient profile, contain bioactive compounds that provide favorable health effects [32]. Bioactive compounds are mainly secondary plant metabolites that are synthesized by the plants to enable them to defend themselves from predators [33]. In legumes, bioactive compounds and other compounds are present as minor components and include polyphenols, saponins, enzymes inhibitors, and phytates (Figure 1). The bioactive compounds that are present in legumes provide favorable effects on health [34]. Consequent to this, legumes can be considered as “functional foods”, due to the presence of nutrients and bioactive compounds. Examples of bioactive compounds present as minor components of legumes and their potential health-promoting properties have been highlighted (Figure 1). 

## 2. Health Effects Associated with Legume Consumption

### 2.1. Effect on Cardiovascular Diseases

Recently, several systematic reviews and meta-analyses of prospective studies have established an inverse association between increased legume intake and cardiovascular (CVD) risk and its disease outcomes [35,36]. For example, increased consumer intake of legumes (≥50 g/day) was associated with a lower risk of coronary heart disease [23] and strokes [36]. CVD, A 6% decrease in CVD risk was associated with increased legume consumption [35]. The plausible reason associated with the improved CVD and coronary artery disease (CAD) outcomes following increased legume consumption could be attributed to the modulation of total cholesterol and low-density lipoprotein cholesterol (LDL-C) blood levels [37], which represent well known risk factors for CVD and CAD. The mechanisms of protection can be explained since fibers, particularly the soluble type from legumes, prevent bile acid recycling as bile acid is removed from the body through feces [38]. This subsequently increases the rate of cholesterol conversion into bile acids and results overall in lower cholesterol concentrations in the body. The second mechanism could involve the role of saponins in the legumes on the modulation of cholesterol levels. Saponins are secondary plant metabolites produced by legumes to provide defense against predatory attacks. Due to their intrinsic presence in legumes, upon legume consumption saponins are hydrolyzed by intestinal bacteria to form an insoluble complex. Saponin complexes interact with cholesterol molecules, impairing the formation of micelles, inhibiting lipase enzyme, and chelating bile acids with subsequent reduction in circulating cholesterol concentration through increased cholesterol conversion into bile acids [39]. Another mechanism could involve the production of short-chain fatty acids, including propionic and butyric acid, as a result of the fermentation of resistant starches from legume fibers [40]. Short-chain fatty acids (SCFAs) have been shown to inhibit hepatic cholesterol synthesis in animal models [41]. A relatively lower production of trimethylamine N-oxide (TMAO) by the gut microbiota following legume consumption was observed, as earlier reported, in comparison to animal-based protein sources deriving from fish or red meat [42,43]. The presence of trimethylamine N-oxide (TMAO) levels in the blood and urine has been established as an objective biomarker associated with increased risk of heart disease. The presence of specific concentrations of micronutrients, including low sodium, high magnesium, and potassium in legumes, may contribute to the reduction of arterial hypertension risk [44]. Legumes are low glycemic index foods with potential insulin-sensitizing effects [45,46,47]. Insulin inhibits the release of free fatty acids from adipose tissue, and this makes legumes impact positively on VLDL and LDL levels [48,49]. 

### 2.2. Effects on Diabetes Risk

Legumes, due to their high fiber and resistant starch content, are effective in controlling postprandial glucose levels and insulin response when compared to other carbohydrate-containing foods [50]. Fiber increases satiety and decreases the absorption efficiency of fats and carbohydrates [51]. Soluble fiber reduces peak blood glucose because it increases the viscosity of the intestinal contents and hinders the absorption of monosaccharides [52], while insoluble dietary fiber modulates the release of gastric hormones and causes delayed absorption of monosaccharides [53].

Dietary plans involving the use of legumes in the medium to long term are desirable in individuals with type 2 diabetes mellitus due to their efficacy in controlling glycemic markers; however, in non-diabetic individuals, individuals with type 1 diabetes, and individuals with prediabetes, the evidence is still limited or contradictory and requires further long-term randomized controlled trials [54,55]. It has been observed that regular consumption of legumes may play an important role in reducing the risks associated with type 2 diabetes mellitus. In a recent systematic review of randomized, controlled trials [56] of individuals with and without diabetes, statistically significant effects were observed in the groups of subjects consuming diets high in legumes with regard to reductions in fasting blood glucose, glycosylated hemoglobin, and blood glucose two hours after a meal. In subjects with type 1 diabetes, the only statistically significant effect found was the decrease in blood glucose two hours after the meal. Improvements in glycemic control were consistently observed on the legume diet in diabetic individuals in several studies identified by this review. Diets containing legumes have also been associated with lower postprandial insulin levels [57]. Several studies [58,59] have also evaluated the effects of legume consumption on the risk of gestational diabetes. Pregnant women may benefit from a legume-based diet to prevent gestational diabetes [58]. Women who adopted a plant-based diet during pregnancy had a significantly reduced risk of developing gestational diabetes compared to women who followed a standard diet [59]. 

### 2.3. Effects on Overweight and Obesity

Due to their particular nutritional characteristics (high fiber and protein content, low glycemic index, low fat content), legumes can be considered a useful food to treat and prevent obesity and being overweight [60]. There are several mechanisms that can explain the weight loss associated with legumes. Some of their satiating properties can be attributed to the presence of fiber, which increases chewing time, delays gastric emptying, inhibits food intake, and stimulates early satiety signals. In addition, soluble fiber in the gastrointestinal lumen forms viscous gels that slow the passage of food through the digestive tract, contributing to the feeling of fullness [61]. The high protein content stimulates the secretion of the hormones cholecystokinin and GLP-1, which contribute to increased satiety [62]. The low glycemic index regulates the secretion of insulin and the blood concentration of glucose, thereby controlling the desire to eat. In addition, legumes contribute to weight loss through the reduced bioavailability of nutrients. High-fiber diets contribute to reduction in fat and protein absorption by reducing the physical contact of nutrients with the intestinal villi [61], by limiting intestinal absorption of nutrients through the formation of viscous gels due to soluble fiber, and by increasing intestinal transit speed due to insoluble fiber [63]. Another characteristic that contributes to making legumes appropriate aliment in weight loss diets is the quality of the carbohydrates: these foods contain a high quantity of amylose-type starch which, after cooking, undergoes the process of retrogradation and becomes more resistant to digestion, compared to amylopectin-type starch [64]. All these mechanisms involving fiber and resistant starches act synergistically to increase satiety, manage body weight, reduce glycemic response, and improve insulin sensitivity [65]. Cell-wall polysaccharides, oligosaccharides (raffinose, stachyose, and verbascose), and resistant starch are the main non-digestible carbohydrates in legumes that are fermented by intestinal bacteria into SCFAs, which are able to influence lipid metabolism and promote fat oxidation and energy expenditure [66].

### 2.4. Effects on Certain Types of Cancer

Some studies have linked the consumption of legumes with a protective effect on certain types of cancer, such as colon, prostate, and breast cancer [67]. According to the World Cancer Research Fund/American Institute for Cancer Research [68], the high amounts of vitamins and minerals in legumes have been suggested to reduce the risk of developing cancer. In vivo studies with rats, in which pre-neoplastic lesions were caused by azoxymethane, showed that eating beans was associated with a lower risk of colon cancer [69]. Similar studies have been conducted to evaluate the effect of bean consumption on induced breast carcinogenesis in rats [70,71]. In the groups that consumed beans, there were dose-dependent reductions in the incidences of breast cancer compared to the control group. In addition, a dose-dependent reduction in the blood concentration of glucose, IGF1 (insulin-like growth factor), C-reactive protein, IL6, and increased apoptosis in mammary adenocarcinoma cells were found in the treated group. 

The non-digestible fraction (resistant starch, insoluble fiber, and oligosaccharides) of many types of common beans possesses anti-proliferative activity and is able to induce apoptosis in colon cancer cells [34]. There are several micronutrients that contribute to the possible anti-cancer effect of legumes: trace elements such as zinc, which is associated with a reduction in oxidative stress and improved immune system functioning, and selenium which, due to its ability to inhibit the development of tumor cells in mouse models of cancer, could play a role in the prevention of breast, esophageal, and stomach cancer [43]. Saponins, protease inhibitors, phytates, and tannins appear to have anticarcinogenic activity due to their antioxidant and regulatory action on cell proliferation [72,73]. Protease inhibitors slow down the rate at which cancer cells divide and stop proteases, which kill nearby cells, from being released. Specifically, black beans contain a trypsin inhibitor that possesses anti-proliferative activity in vitro [74].

### 2.5. Prebiotic Potential

The high amounts of non-digestible oligosaccharides (raffinose, stachyose, and verbascose), resistant starch, and other non-starch polysaccharides in legumes contribute to the formation of SCFAs in the large intestine. This confers to these foods “prebiotic potential” by changing the composition of the intestinal microbiota and increasing the growth of Bifidobacteria. Among the short-chain fatty acids, in particular propionate, has been shown to decrease cholesterol levels, reduce endogenous fatty acid synthesis, and promote satiety [66]. SCFAs can influence enterocyte growth because butyrate is their main source of energy and appears to protect against cancer with anti-inflammatory and anti-tumor effects on colon cancer cells, and also affects mineral availability because a lower pH in the colon allows calcium and magnesium to dissolve more easily [75]. To figure out if a diet high in legumes can change the gut bacteria in a good way, more research needs to be carried out to investigate their prebiotic potential.

### 2.6. Oxidative Stress and Inflammation

Since oxidative stress and inflammation are recognized as playing crucial roles in the onset of age-related and chronic diseases, they may be fruitful dietary targets for preventing illness [76]. Legumes contain polyphenols in the form of phenolic acids and flavonoids, which have anti-inflammatory and antioxidant properties and protect tissues from oxidative stress. Flavonoids counteract free radicals, endothelial dysfunction, and platelet aggregation [77]. The intake of legumes four times a week in a nutritional plan against obesity with a moderately low-calorie diet resulted in a decrease in lipid-associated oxidative stress with a decrease in lipid peroxidation biomarkers, such as oxidized LDL and malonylaldehyde, and also in C-reactive protein, independent of weight loss [78].

## 3. Barriers towards the Consumption of Legumes

Several factors impact consumption of food crops including legumes. In the case of legumes, despite their rich nutritional and health-promoting properties, analysis of their consumption using data from the Global Dietary Database (GDD 2018) in June 2022 showed a general global decline in beans and legume consumption [22]. The decline in consumption was more prominent in several high-income countries including the United States of America and Canada. In that same study, the authors compared the national legumes and beans consumption with the 50 g/day target which has been established in several meta-analyses to be the intake required to reduce mortality and morbidity outcomes [22]. The general decline observed was attributed to factors including the lack of clarity regarding the quantity and serving size of beans and other legume products despite the availability of national food-based dietary guidelines. Additionally, most of the food-based dietary guidelines group legumes with seeds, vegetables, and starchy staples. Consequently, due to the aforementioned challenges in Australia, consumers declared that the use of the statement “each day, consume at least one serving of legumes either as a serve of vegetables or as an alternative to meat” was enough to compel them to increase their consumption of legumes [25]. 

Despite the factors highlighted above, other underpinning barriers towards legume consumption include food neophobic tendencies, digestibility and health-related concerns, and socio-economic reasons and food taboos.

### 3.1. Phychosocial and Socio-Economic Reasons

#### 3.1.1. Food Neophobia Tendencies

Food neophobia is a dietary behavior associated with the lack of desire for a consumer to taste new food products or other unfamiliar food products [79]. The drivers of this could be attributed to perceived unappealing organoleptic attributes of the food, health complications due to allergy-triggering tendencies, and traditional beliefs. Legumes exist in different forms apart from the other well-known types including chickpea, soybean, cowpea, and lentils. Consumers used to one type of legume are reluctant to try new ones. In Ghana, for example, although there are now efforts to promote the consumption of other uncommon legumes such as the broad bean types, consumers appear to have an apathy towards consuming these legume types. Karaağaç et al. [79] suggested that educational programs and activities related to food in order to promote positive attitudes and experiences towards food could be an efficient and promising tool to overcome food neophobia tendencies and could subsequently result in improved consumer interest in tasting familiar and unfamiliar foods.

#### 3.1.2. Food Taboos 

Food taboo is another important factor that promotes the lack of interest in legume consumption. It results in nutritional deficiencies and adversely affects the nutritional status of consumers. Meyer-Rochow et al. [80] have extensively reviewed food taboos amongst different religious and geographical orientations. Food taboos connote restrictions toward the consumption of certain food types, driven mainly by religious doctrines and traditional beliefs [80]. This practice is common in several low- and middle-income countries, including in Africa. For example, recently, a cross-sectional study involving 332 pregnant women was conducted in Ethiopia to investigate food taboos and other related misperceptions [81]. The authors reported that 45.7% of the pregnant women reported the consumption of legumes as a food taboo. The women reported that factors including development of abdominal cramps, prolongation of labor with subsequent trigger of pain sensations, and possible abortion induction were the predominant reasons to avoid beans and chickpea intake.

#### 3.1.3. Socio-Economic Factors

Over the last thirty years, the revolution in eating habits in industrialized countries has led to the need and desire, for economic and social reasons, to spend less time on food preparation in the kitchen. In fact, most grain legumes, when not purchased already cooked, require long processing and cooking times that may discourage their use. Changes in lifestyles, less time available for cooking, the greater availability of affordable processed foods that require less time to prepare, and greater commercial pressure have led in some southern European countries (Italy, Greece, Portugal) to a gradual abandonment of the so-called Mediterranean diet, based on the intake of products such as vegetables, fruit, whole cereals, legumes, oil seeds, dried fruit, extra virgin olive oil, fish, and wine, and to an increase in the consumption of processed and refined foods and sugary drinks [82]. The same trend is being observed in many Latin American countries, traditionally large consumers of pulses, where rapid urbanization and socio-economic changes have caused changes in their eating habits [83]. 

### 3.2. Digestibility and Health-Related Concerns

Legumes, despite their rich nutritional profile, contain other compounds including non-digestible carbohydrates, such as alpha-galactosides and anti-nutrients that could impair human health, as their presence in legumes can cause nutritional deficiencies by limiting protein digestibility, the bioavailability of minerals and vitamins, and altering the intestinal epithelium [84]. Oligosaccharides, such as raffinose, stachyose, and verbascose, which are a class of carbohydrates referred to as alpha-galactosides, are intrinsically present in legumes although their concentration varies depending on the species of legume consumed. The oligosaccharides cannot be hydrolyzed and absorbed by the human digestive system due to the lack of the enzyme alpha-galactosidase (Table 2). 

Consequently, these carbohydrates reach the colon undigested, where they undergo an anaerobic fermentation process by the intestinal microbiota, resulting in the formation of SCFAs and gases (hydrogen, carbon dioxide, and methane) responsible for bloating, distension, abdominal pain, and diarrhea [85]. Flatulence is one of the factors that make legumes unacceptable in the diets of western countries and discourages their use. A high raffinose content in the diet (>6.7%) has osmotic effects in the intestine and contributes to the onset of diarrhea, along with excessive fermentation [75]. The osmotic pressure imbalance generated in the small intestine by oligosaccharides of the raffinose family reduces its absorption capacity. It has been established through in vivo studies that the presence of lupin oligosaccharides in the intestinal mucosa reduced the efficiency of protein digestibility. However, the extraction/elimination of these compounds from lupin resulted in significant increase in all lupin amino acids using pig models [75]. Furthermore, these oligosaccharides appear to be implicated in the increasing susceptibility to chronic gut diseases such as Crohn’s disease and, especially, irritable bowel syndrome, in which they are considered irritants [86]. Finally, studies have shown that alpha-galactosides are responsible for decreasing the net energy content of the diet [75].

Anti-nutrients are another class of compounds in legumes that impair the digestibility of proteins, and the bioavailability and bioaccessibility of minerals and vitamins [84]. This subsequently results in nutritional deficiencies in the body. They also alter the intestinal epithelium [84]. Most of the anti-nutritional factors have important physiological functions and are indispensable for the plant’s defense mechanisms; therefore, it is not possible to remove them from seeds during cultivation, as this would be incompatible with agronomic requirements. Examples of some commonly reported anti-nutrient compounds in legumes include protease inhibitors, lectins, phytic acid, oxalate, and saponins. Toxic effects associated with common anti-nutrient compounds in legumes are presented below (Table 3). 

Protease inhibitors reduce trypsin and chymotrypsin activity, resulting in impaired protein digestion [88]. They have also been associated with growth inhibition and pancreatic hypertrophy in certain experimental animals [89]. Alpha-amylase inhibitors have also been associated with an increase in pancreatic hypertrophy [90]. They form a complex with amylase, reducing its activity and resulting in reduced starch digestion [64].

Lectins are present in many bean species; they have various functions in the plant, such as physiological regulation, defense against microorganisms, protein storage, carbohydrate transport, and recognition of nitrogen-fixing bacteria of the genus Rhizobium [69]. They can form specific bonds with sugars and enterocytes, modifying their ability to absorb nutrients; they also have an agglutinating action on red blood cells, depending on the type of legume and the type of heat treatment it undergoes [90]. They are sensitive to heat treatment; improperly cooked beans can be toxic to humans, causing nausea, vomiting, diarrhea, and bloating, likely due to incomplete denaturation of the lectin which can withstand moderately high temperatures [90].

Phytic acid (inositol-hexaphosphoric acid) represents the main phosphorous reserve in the plant and has the ability to chelate multivalent metal ions, particularly iron, calcium, and zinc, with which it forms insoluble complexes that are more difficult to digest and therefore less available for intestinal absorption [90]. Oxalates are found in most legumes in the form of potassium or calcium salts and, like phytates, are capable of decreasing the bioavailability of mineral salts, particularly calcium. Food-derived oxalate is also involved in the formation of calcium oxalate kidney stones [90].

The main phenolic compounds found in legumes are tannins, phenolic acids, and flavonoids, and the species that contain the most polyphenols are the dark ones (red bean and black bean) [90]. Tannins have reactive groups that can form stable complexes with proteins and other macromolecules, especially during cooking. They are able to inhibit the activity of proteolytic enzymes, leading to a reduction in protein digestibility; they precipitate salivary proteins, leading to the characteristic astringent taste. They also change the intestinal pH, altering the mucosa and reducing micronutrient absorption. Finally, they are able to form complexes with minerals and trace elements, reducing their bioavailability [90].

Saponins are a group of glucosides known for their property of reducing surface tension and forming stable foams in aqueous solutions. Among legumes, beans are particularly rich in such compounds and may contain various types of saponins. At high concentrations, they impart a bitter and astringent taste to the food, which limits its consumption in human nutrition [69]. Their anti-nutrient effect is due to their haemolytic activity caused by interactions with the cholesterol present in red blood cell membranes; due to their ability to interact and form mixed micelles with bile acids and cholesterol, they have been studied for their cholesterol-lowering action, but their long-term toxic effects are still unknown [91].

The pyrimidine glycosides present in the cotyledons of fava beans (vicine and convicine) are responsible for favism: a condition in which individuals with congenital glucose-6-phosphate-dehydrogenase deficiency suffer acute haemolytic anaemia, as the protective effect of the enzyme is lost. These glucosides are stable upon cooking. Legumes that contain vicine and convicine (fava beans in primis, but other legumes may also contain small amounts) should be banned from the diets of those subjects with favism [90]. Cyanogenic glycosides, although present in very small quantities in some bean species (especially black lima bean varieties) and chickpeas, can induce respiratory distress if eaten in large quantities, as after enzymatic hydrolysis by an endogenous glucosidase they release hydrogen cyanide and acetone [92]. Lathyrogens are amino acid derivatives with toxicity, present in chickling vetch seeds. They can cause lathyrism: skeletal deformities, muscular rigidity, and paralysis [90].

Most of the antinutritional factors (enzyme inhibitors, phytic acid, polyphenols, and saponins) in the right quantities have been shown to have beneficial properties, which is why it is important not to eliminate them completely from the diet but to keep intake below toxic levels [32].

## 4. Strategies to Promote Increased Consumption of Legumes

There are several drivers of food intake, including those of legumes, and these drivers mostly tend to take psychological, social, and psychosocial orientation. Promoting increased consumption of legumes requires that the barriers towards the consumption of legumes are addressed. An important approach that could be exploited involves the application of appropriate food processing technologies on legumes, which is essential in addressing digestibility and health-related concerns associated with legume consumption. This could lead to the production of legumes with a reduced content of non-digestible carbohydrates and anti-nutritional factors, and with reduced cooking time (Figure 2). 

Other strategies include culinary approach, such as food reformulation leading to the development of innovative, functional, and nutrient-dense new legumes and legume-based products with appealing organoleptic attributes intended for targeted consumer groups (e.g., vegans, vegetarians, coeliacs, sportspeople, lovers of healthy food) that satisfy the growing demand for healthy and easy-to-prepare foods; political and economic strategies; the promotion of effective communication campaigns; and food education projects that involve all those involved in the food chain and health professionals that overturn and modernize the image of legumes.

### 4.1. Processing Methods to Reduce Alpha-Galactosides

Examples of processing methods that could be exploited to reduce alpha-galactosides include extractive methods (soaking and baking), thermal degradation methods (autoclaving and extrusion), enzymatic degradation methods (germination, fermentation, enzymatic degradation), and degradation through gamma rays [85]. The preceding sections below will provide insight into the various methods that can be used in order to reduce alpha-galactosides.

#### 4.1.1. Extractive Methods Application for Legume Processing 

This method involves the use of a 50% hydroalcoholic solution at 40 °C for the extraction of alpha-galactosides from various legume seeds [93]. Subsequently, functional legume extracts are obtained, and the extracted alpha-galactosides can be purified and used for their prebiotic activities [75]. 

#### 4.1.2. Soaking of Legumes 

This is the most traditional, simple, and economical method based on the water solubility of alpha-galactosides. The process involves dried pulses being placed in a container of water and left to rehydrate for a variable period of time, usually overnight. Losses of these substances are partly due to extraction by osmosis in the soaking water and partly because metabolic processes take place within the soaked seeds, resulting in the further degradation of the alpha-galactosides and the subsequent release of monosaccharides and disaccharides into the water [85]. The effectiveness of this technique is influenced by various factors, such as the type and variety of legume, the presence of salts in the water, the temperature of the water, the soaking time, and the legume-to-water ratio [94]. 

#### 4.1.3. Cooking of Legumes 

The reduction of oligosaccharides in legumes occurs mainly through the mechanism of diffusion [95]. Studies, in general, indicate a direct relationship between increased cooking time and increased oligosaccharide reduction. Specifically, a significant decrease in the concentration of raffinose, stachyose, and verbascose in common beans by 39.2%, 48.1%, and 40.4%, respectively, was recorded after half an hour of cooking [29]. Similar results were reported following the cooking of an eye bean meal [96] and a cajano bean meal [97]. The reduction in the concentration of alpha-galactosides is even more prominent in legumes when soaking is employed as a pre-treatment method prior to cooking. In fact, it has been reported that in chickpeas cooked without soaking, the stachyose content increased by 133%, and the verbascose content in peas by 282%; similar trends were also observed with some bean species [85]. The plausible mechanism underpinning this observation involves a quantity of oligosaccharides being intrinsically bound to the proteins in the legume seed. As a result, the oligosaccharides are not detectable. However, during cooking, the increased heating temperatures result in the denaturation of the bound proteins, resulting in the free release of the oligosaccharides inside the seed which are subsequently quantified [98].

#### 4.1.4. Autoclaving Application on Legume Processing 

Pedrosa et al. [99] have extensively carried out a systematic review associated with the application of autoclaving as a processing method on the bioactive composition of legumes. During autoclaving, the legume is placed in a pressure chamber saturated with steam at a temperature of 121 °C and a pressure of 1.8 to 2.0 bar, and cooking is carried out for a shorter duration. The induced pressure forces water inside the seed, allowing a more efficient extraction of oligosaccharides than simple cooking. Autoclaving, on average, removes between 49.20 and 56.00% of the total alpha-galactosides from the legumes, depending on the species of legume, compared to 41.5–47.6% with traditional cooking when using raw legumes. The same trend is observed with pulses soaked prior to treatment: a 56.8–70.0% removal of total alpha-galactosides with autoclaving, compared to 49.7–64.0% with traditional cooking. Furthermore, the time required for these results is longer with traditional methods, 60 min, than with the autoclave method, which only requires 20 min. The effects of adding substances to the water used for soaking were also examined, and the most promising results were recorded with the addition of, in this order: tamarind pulp, an alkalinizing substance, and citric acid, compared to water alone. Finally, it was found that the most favorable bean-to-water ratio for the removal of alpha-galactosides in autoclaving is 1:10 *w*/*v* [85].

#### 4.1.5. Extrusion

Extrusion is a process that consists of forcing a semi-solid mass product (in this case prepared from legume flour) through a shaped mold. During extrusion, the food matrix is subjected, for a short time, to a high temperature (100–150 °C) and subjected to shear and pressure forces; these stresses trigger modifications from a chemical, micro-, and macro-structural point of view, which lead to a mixing and homogenization of the food mixture; a product with different structural and nutritional characteristics compared to the starting food is obtained [100]. Various studies on different legume flours under different conditions have shown heterogeneous results in the reduction efficiency of alpha-galactosides [85]. The most significant results were obtained using legume flours formulated with the addition of starch, fiber and flavorings [101]. The use of low extrusion temperatures (85 °C) allows the nutritional properties of the legumes to be preserved at the expense of a significant reduction in raffinose but not in stachyose. At 142 °C, on the other hand, there is a significant reduction in total alpha-galactosides, amounting to 28% [85].

#### 4.1.6. Enzymatic Degradation of Alpha-Galactosides 

This method includes processes that use the enzyme alpha-galactosidase in order to break down oligosaccharides into sugar molecules and galactose residues. For example, during germination, the activity of the enzyme alpha-galactosidase of legumes becomes activated. This allows the complete removal of raffinose family oligosaccharides (RFOs), ciceritol, stachyose, and verbascose due to the breakdown of alpha-galactosides, a substrate of the enzyme. The amount of alpha-galactoside removed is dependent on the germination conditions available [75]. Germination occurs after soaking the seed and can take place in the dark or in the light; in the dark, the process takes a shorter time [85].

Fermentation uses microorganisms (bacteria and yeasts) that produce the enzyme that degrades alpha-galactosides. This process can occur spontaneously when using the microorganisms naturally present in the seed (natural fermentation) [102] or through the use of microbial cultures, usually consisting of lactic acid bacteria or molds (induced fermentation). In this case, the results differ depending on the Lactobacillus strain used. Natural fermentation seems to provide the highest reduction of alpha-galactosidase when fermenting various legumes [85].

With enzyme treatment, the enzyme alpha-galactosidase is synthesized and added directly to the whole bean or flour, followed by an incubation phase to increase enzyme activity. In one study, the purified enzyme obtained from Aspergillus niger was used to treat a Cajun bean flour, revealing the complete removal of raffinose, stachyose, and verbascose after treatment [103]. Similar results were obtained on the same type of legume in another study [97]. This type of approach is more effective than soaking and cooking but can alter the sensory quality of the legume, modifying its flavor, altering the perception of bitterness and astringency, and generating unexpected flavors [104].

#### 4.1.7. Gamma Ray Application

Gamma rays hydrolyze alpha-galactosides because they cause the cleavage of the alpha-1,6 bond between galactose molecules, leading to the formation of glucose, galactose, and melibiose. This method does not adversely affect the sensory and other functional properties of the food. Studies that have been carried out on this subject are still scarce because consumers probably have a negative perception of irradiated foods [85].

#### 4.1.8. Genetic Manipulation

Alpha-galactosides play an important functional role as reserve carbohydrates, for transport mechanisms and to enable the plant to resist low-temperatures; therefore, all attempts at genetic manipulation to reduce these oligosaccharides must allow the plant to remain viable until harvest [75]. In addition to using regulatory genes or genes that encode for alpha-galactosidases, a further application would be to obtain genes that encode enzymes that are thermostable, with optimal temperatures around 100 °C, using hyperthermophilic bacteria. These enzymes would only be activated after harvest, through a process of heating the seeds, causing the degradation of the alpha-galactosides but leaving them intact and usable by the plant throughout its life cycle [75].

## 5. Strategies to Adopt towards Reduction of Anti-Nutritional Factors in Legumes

Anti-nutritional factors are compounds present in legumes that impair the bioavailability of essential nutrients, leading to potential nutritional deficiencies including hidden hunger. The elimination or reduction of anti-nutritional factors can be achieved through domestic methods (soaking, cooking), or through technological treatments that are often used in combination. The most frequently used methods include dry and wet heat treatment, extrusion cooking, steaming, soaking, sprouting/germination, fermentation, dehulling, and enzymatic treatment. The chemical and physical characteristics of the anti-nutritional factors determine the most suitable treatment for reducing the unwanted substance [84]. Sprouting can be used to reduce the content of phytates, tannins, and enzyme inhibitors. Hulling, soaking, and cooking are effective methods to reduce the content of high molecular weight tannins; selecting cultivars with low tannin concentrations also results in more digestible legumes [90]. In 2016, a review article examined the optimum conditions associated with soaking and sprouting as essential for reducing the concentrations of phytates, tannins, and polyphenols in chickpeas and common beans [105]. The authors reported that the soaking and sprouting/germination of the legumes led to significant reductions in the concentration of anti-nutrients, and this varied depending on the duration of soaking and sprouting, as well as the legume type. The highest phytate reduction for chickpeas and beans was achieved with a germination time of 96 h. The best results for tannin reduction were observed with a 48 h germination for chickpeas and a 4 h soaking for beans. No proportionality was observed between tannin reduction and the amount of water used for soaking, nor was tannin reduction proportional to soaking time. Although studies indicate different percentages of anti-nutrient reduction, the usefulness of these two techniques in improving the nutritional quality of chickpeas and beans is evident.

Enzymatic degradation may also be an effective means to reduce the concentration of anti-nutrients. For example, an enzyme capable of degrading phytates has been isolated and characterized from broad bean seeds and used in processes to remove phytates from legumes. Furthermore, enzymatic treatment or fermentation of legume proteins using microbial proteases has been suggested as an effective procedure to increase digestibility, decrease allergenic potential, and improve the functional properties of legumes [106]. Even though several heat-treatment methods are effective in reducing the concentration of most anti-nutrients in legumes, certain heat-stable anti-nutrients are present and require the application of high temperatures to obtain their removal. Treatment with very high temperatures results in structural changes in proteins that alter their digestibility and may interact with other nutrients; consequently, this is not always advisable [84].

### 5.1. Techniques to Reduce Cooking Time

Considering the food environment and factors that impact on consumers’ energy-dense, nutrient-poor, and convenient foods as preferential choice, it has been observed that the primary driver includes convenience and the lack of time required for cooking healthier meals. The lack of time is determined by factors including longer working hours spent at the workplace, which is relevant in both industrialized countries with hectic lifestyles and developing countries with limited resources for meal preparation. Regarding the use of legumes, the lengthy preparation and cooking times pose significant challenges that restrict their use as ingredients in diets. The application of traditional approaches, such as soaking and dehulling, facilitates water absorption into the cotyledon, resulting in improved texture and softness of the legume seed for cooking. However, this process also reduces anti-nutrient factors such as polyphenols and tannins, as well as raffinose and stachyose, but it also leads to a reduction in fiber, iron, calcium, and niacin.

Various processing technologies can be used as pre-treatments to obtain pre-cooked pulses that require less cooking time. These technologies include gamma irradiation, pulse electric field, microwave heating, and infrared heating. During gamma irradiation of legumes, ionizing radiation is produced which penetrates the cell walls of the legume seed, resulting in structural modifications of the legumes [107]. This subsequently leads to a reduction in the cooking time required for the legume cooking. 

Pulse electric field is a non-thermal processing method that could be applied to legumes as a pre-treatment method to reduce cooking time. Briefly, during the pulse electric field treatment, legumes are placed between two electrodes and high-voltage pulses of electric field are passed through them for a short duration (ranging from nanoseconds to milliseconds) [108]. The mechanism involves the absorption of energy by the polar groups of amino acids, which are the building blocks of proteins, leading to the generation of free radicals [109]. The produced free radicals interfere with the bonds that exist between proteins resulting in structural and conformational changes in the proteins. Devkota et al. [110] recently investigated the effect of thermal and pulsed electric field (PEF)-assisted hydration on the hydration behavior, mass loss, and leaching of bioactive compounds in two common beans. The authors reported that the application of pulse electric field resulted in a threefold reduction in the time required for hydrating legumes, especially the slow-hydration types, compared to the soaking process at 45 °C [110]. 

Microwave heating is a technique that can also be used to enhance flavor, reduce protease inhibitors, gelatinize starch, and denature red bean proteins resulting in instant flour with higher water absorption, increased fat-binding capacity, greater viscosity, and foaming capacity [111]. Infrared heating is an emerging technology that enables very high temperatures to be reached in a short time, leading to increased water absorption during soaking and cooking, reduced cooking time, and improved sensory qualities including color, flavor, and texture. It can also produce instant flours with modified functionality and enhanced fat absorption capacity [112]. Further studies and investigations are needed to assess the consequences and mechanisms through which infrared heating may affect the nutritional properties of legumes [112].

### 5.2. Alternative Use of Legumes and Their Derivatives 

In recent times, factors including increased diet-related non-communicable diseases and climate change with its attendant food insecurity challenges have caused a surge in consumers’ interest in foods that are produced from sustainably sourced food raw materials and have health-promoting benefits. Legumes share these traits and, thus, their utilization in either reformulated or newly developed food products with appealing organoleptic properties and labelled using a front-of-pack approach with the message of clean label could resonate with consumers drivers for food purchasing. Consequent to these, the prospects of applying technological processes for the production of legume-incorporated foods in extruded products such as bars, croquettes, puffs, galettes, instant flours, soups, bread and other bakery products, pasta with a higher protein content, vegetable or vegan burgers and meatballs, meat substitutes, emulsions, and drinks would be highlighted (Figure 3). 

Enrichment of legumes in these products could also pose three major challenges including technological drawbacks, nutritional limitations related to amino acid composition and the presence of anti-nutritional factors, and sensory alterations related to organoleptic characteristics.

### 5.3. Ready-to-Use Snacks, Breakfast Cereals and Meat Alternatives

Extrusion technology is an important technique that can be employed for the production of various foods, such as crispy snacks (bars, galettes), creamy puffs, breakfast cereals, instant soups and meat substitutes [112]. In recent times, the prospect of incorporating legumes into cereal-based products through the application of extrusion technology has been explored. The rationale primarily has been to improve the nutritional density, especially in relation to providing an excellent source of protein, starches, fiber, vitamins, and minerals, and could be important in contexts where, for economic, dietetic, ethical, and religious reasons, access to animal protein is limited. Extrusion technology requires the use of an extruder for the development of food products. During extrusion, food materials are mixed, kneaded into a moldable form, and forced through a small orifice, called a die, under the action of high pressures generated by one or two screws that compress the dough, cut it and transform it. Extrusion makes it possible to transform hard, unpalatable materials into lighter, more consumer-acceptable forms [113]. It has been reported that the most appropriate amount of legume meal to be incorporated into extruded cereal-based foods in order to balance the nutritional characteristics and obtain a product with good structural and sensory properties should not exceed 30%. This allows for the attainment of food products with excellent amino acid composition and appealing physicochemical and textural properties such as lightness, crunchiness, and hardness, compared to their non-legume-enriched products [113]. Extruded legume powders can also be used to fortify baby foods to be reconstituted with water. For these foods, higher percentages of legumes can be added without compromising their sensory properties: mixing maize flour with chickpea flour at percentages that can reach over 70% provides an excellent balance from a nutritional point of view with a good content of essential amino acids. Increasing the extrusion temperature can lead to an increase in starch digestibility of up to 90% because there is an improvement in starch gelatinization; in addition, there is shear-induced cleavage of amylose and amylopectin with the formation of smaller, more digestible fragments (dextrinization). By adjusting the processing parameters, the digestibility of starch can be modulated: foods intended for infants should have a high digestibility, while other products intended for obese people should contain less-digestible material. Conversely, an increase in temperature may lead to a loss of lysine. Protein digestibility also increases with extrusion cooking due to loss of protease inhibitors, phytates, tannins, and polyphenols, all thermolabile components that reduce protein digestibility.

Meat alternatives have become an area of interest to consumers in recent times. This is driven largely by health concerns and concerns for environmental sustainability. Vegetarians, for example, are one group of consumers with increased interest in meat alternatives. Recent advancement in food product development has led to the development of plant-based food products that have textural attributes that mimic meat products. The technological challenge involved is largely attributed to the ability of food manufacturers to develop meat alternatives with improved structural attributes that could mimic those of animal-based meats and with sensorial attributes that mimic those of animal-based meats [114]. The use of extrusion technology has shown promise in delivering legume-based meat alternatives that can compare well in terms of structural and sensorial attributes.

### 5.4. Bakery Products

Bakery products, especially bread, remain one of the commonly consumed staple foods globally [115]. Consequently, bakery products could be targeted as promising nutrient delivery media for including nutrients from legumes. Recently, there has been a surge in consumer interest for functional food products including bakery products [116]. This has been predominantly attributed to the health-promoting benefits associated with functional food consumption [116]. White bread and other bakery products formulated with white wheat flour are inferior in nutrients, especially in fiber, due to their refined nature. Increased consumption of these products is a risk factor for the development of type 2 diabetes as they are high glycemic index foods [117]. Legume flour is nutrient-dense and its enrichment with white wheat flour could enhance its nutritional value and improve its health-promoting properties. Common legumes used for bread enrichment include white kidney bean, chickpeas, and lupin [118]. Health-promoting benefits reported, for example, for bread enriched with white kidney bean, lupin, and chickpea flours include lower postprandial glucose and insulin release [118]. 

In bread-making, wheat flour is essential because it contributes towards gluten enrichment, which is a protein essential to trap carbon dioxide produced during the leavening process. Since gluten has a fundamental function in the structural development of bread, even the partial replacement of wheat flour with legume flour presents a challenge due to the absence of gluten, resulting in a bread with a lower consistency and a less porous crumb structure. The most acceptable results are obtained using percentages of legume flour below 15% [119]. Furthermore, the enrichment of bread with legume flour contributes to an increase in Maillard reaction, which consequently leads to the darkening of the crumb. This may lower the acceptance of taste and other organoleptic attributes of legume-enriched breads in general [120]. It is important, therefore, to ensure that the quantity of legume flour used for enrichment is ideal to allow product acceptability. The best sensory results in terms of appearance, taste, and color are obtained with the addition of up to 10% of legume flour (especially peas), while larger proportions lead to a deterioration of the product’s sensory profile [121].

In the development of biscuits, the formation of a gas-tight network of gluten during leavening and baking is not indispensable; therefore, the use of legumes is very suitable in formulations for coeliacs because gluten does not play a key role in the structure of these foods. Moreover, the increased firmness and hardness for biscuits and crackers could be a positive aspect. Crackers lend themselves well to reformulation with legume flour. Using 30 percent green lentil flour, a product rich in protein and with a higher dietary fiber content than crackers formulated with wheat is obtained; the color is darker, but the texture and taste are comparable to classic products [122]. Gluten-free products can be obtained by completely replacing wheat with flours of chickpeas, green and red lentils, borlotti beans, kidney beans, black beans, or yellow peas, with the organoleptic characteristics appreciated by consumers [123]. To obtain a gluten-free product, mixing it with rice flour and tapioca starch allows the production of food with acceptable taste and texture characteristics [122]. In baking cakes, the results are less satisfactory. Using mixtures containing chickpea or pea flours results in a decrease in volume, an increase in compactness, and a decrease in dough cohesion and fluffiness [124].

### 5.5. Pasta

Various types of pasta prepared with different legume flours (lentils, peas, and chickpeas) are commercially available. These products have a higher protein and fiber content than traditional pasta, a lower carbohydrate content, and a lower glycaemic index. Being gluten-free, they are suitable for the diet of coeliac patients [125]. They are not to be understood as a substitute for traditional pasta due to their different composition, but rather as an alternative approach to increasing the consumption of legumes, or as an indication in low-carbohydrate diets. These products require a shorter cooking time than durum wheat pasta and tend not to keep cooking as well [125]. Generally, pasta is formulated from refined durum wheat due to the presence of gluten from the wheat flour. Upon enriching the pasta with legume flour, the gluten concentration is diluted. A technological challenge that appears is that, when too much of the legume flour is incorporated into the pasta, the legume-enriched pasta may experience increased cooking losses. To overcome this issue, it is important to study the functional properties of the legume-refined wheat flour composite to appreciate the effect of their interaction on their structural properties. Additionally, preliminary studies can be carried out to appreciate the amount of legume flour that would be enough to produce enriched products with desirable attributes. 

### 5.6. Other Products

Legume flours can also be used to prepare emulsions for use as condiments or fortified drinks, prepared by supplementing orange and apple juice with pea protein isolates or with chickpea and lentil flours [125]. In recent years, consumer demand has exploded for vegan and plant-based burgers and meat substitutes, products that are devoid of animal protein but have overlapping chemical characteristics and are intended to provide a product with the same appearance, shape, flavor, and texture. These products can be prepared using soya, gluten, and pulses (chickpeas, peas, lentils, and beans) as protein sources. Among the technologies used for production are high-moisture extrusion and cutting-cell technology which, through mechanical and thermal stress, allows for the alteration of the protein structure of plant origin, resulting in proteins with fibrous structures that closely resemble meat. By modifying the process conditions, products with different elasticity and sponginess characteristics can be obtained that can take on the appearance of hamburgers, minced meat, and sausages [126].

### 5.7. Political and Commercial Strategies

Assuming that legumes have a positive role not only for health but also for the environment, strategies to bring about a change in the consumption of legumes should involve all stakeholders within the food chain. Legume crops have a low environmental impact due to their low carbon and water footprint that has a much smaller impact than cereals and other protein sources (milk, meat, and eggs) [127].

The growing social awareness of the impact of climate change and agricultural practices should be a stimulus for the food industry to offer consumers alternatives that meet the growing demand for innovative products with nutritional benefits and a low environmental impact. Australia has seen a remarkable increase in the consumption of pulses in the last two decades of the last century, from half the consumption of the European Union in 1979, to double in 1999; trade strategies have made a strong contribution to this achievement, demonstrating that change is possible [128]. Increased investment in research and development (for the development of modern, healthy, varied high-quality products and from functional foods to ready meals and traditional foods) encourages consumer demand and stimulates the food industry to invest more in these products, thereby expanding their market and meet the needs of different types of consumers. In marketing and advertising, with modern and effective communication campaigns that highlight the benefits of these foods, as well as support for local policy development by promoting the local origin of products, traceability and environmental sustainability are necessary to achieve this. Educational campaigns supporting the partial substitution of cereals by legumes could strengthen the nutritional density of basic meals and prove effective in countries where dietary patterns are low in quality and include high-energy, low-nutrient-density foods, which are associated with nutrition-related disorders such as increased cardiovascular risk, overweight and obesity, and diabetes [129]. In national dietary guidelines, there is a tendency to bring together foods with similar nutritional and biological characteristics into the same group. When analyzing globally in which food group legumes have been placed, a certain heterogeneity can be observed: in Bulgaria, Canada, and Ireland, they have been included in the group of meat and meat alternatives due to their high protein content; in Australia, Nordic countries, the United Kingdom, the United States they have been included in both the group of meat and meat alternatives and in the group of vegetables; in India, they have been included in the food group that also includes cereals due to their starchy content; in Brazil, Spain, South Africa, and Greece, they are placed in a separate food group [129]. An international group suggested standardizing the portion size of legumes worldwide in the guidelines to 100 g or half a cup (125 mL) of cooked product, and classifying legumes as an independent food group, to promote their consumption [129]. A 100 g portion is a reasonable minimum portion size that can enhance the nutrient density of healthy diets; the nutritional density of 100 g of cooked legumes is also demonstrated when the nutritional composition of legumes is compared to the Institute of Medicine’s Dietary Reference Intake values. Taking adults into consideration, 100 g of legumes provide up to 32%, 28%, 20%, and 17% of the RDA (recommended daily allowance) for folate, iron, phosphorus, and magnesium, respectively. A 100 g portion of pulses also provides up to 15–19% of the RDA for protein, and for RDA values not available, a half-cup of pulses provides 19% to 35% of the adequate intake (AI) for fiber and 9% of the AI for potassium (Table 4). 

This table shows the nutrient content of 100 g dry legumes and their corresponding recommended dietary allowance (RDA) and adequate intake (AI) values. The RDA represents the average daily dietary intake level that meets the nutrient requirements of nearly all healthy individuals in a specific gender and age group. The AI represents an estimate of the amount of a nutrient that is adequate to meet the needs of most healthy individuals within a certain population group. It is important to note that individual nutrient needs may vary depending on factors such as age, sex, and activity level, and the RDA or AI should not be considered strict requirements for nutrient intake. Note: data are modified from [28].

Such a well-defined portion size would harmonize international strategies to communicate the nutritional benefits of legumes based on their important contribution to the nutrient density of healthy diets. The frequency of consuming the standard portion could then be adapted to local needs and kitchen characteristics. 

It would also be important to understand the barriers and opportunities for promotion to increase consumption of pulses. A paper published in 2015 that examined barriers, attitudes, and consumption of lentils in households in Canada reported that 85% of respondents did not consume pulses due to a lack of knowledge and the belief that family members would not like them [126]. Exploring these issues and understanding why some groups of individuals regularly consume pulses would be a lever to encourage their consumption in other groups providing important insights into effective consumption incentive strategies.

### 5.8. Food-Based Dietary Guidelines

The use of the food-based dietary guidelines is a population-based public health approach that could promote increased intake of health-promoting food products, leading towards improved health outcomes. For example, in New Zealand, the use of 5+ A Day promotes increased intake of fruits and vegetables in quantities that allow consumers to meet their fruit and vegetable requirements [130]. The FAO encourages a minimum intake of at least 400 g of vegetables and fruits a day [131]. In communicating these guidelines, the message should be succinct and clear in terms of its serving size and quantity to consume. To promote legume intake, the wording of the message used to promote legume consumption should be clear [25]. Additionally, legumes should be decoupled from other food groups on the food based dietary guidelines. This, coupled with intensified public health campaigns around the health-promoting effects of legumes, could promote increased intakes.

### 5.9. Clean Labelling of Innovative Legume-Enriched Food Products

Food labels impact consumers’ purchasing behavior. For example, the use of front-of-pack labels [132], which clearly communicate the legume ingredients used for the product formulation, could promote consumer trust and its purchase. Even more important is when these innovative new legume-based products are produced without synthetic food additives. Additionally, the display of validated health claims regarding the products could also promote consumers’ purchase of the products. For example, the display of “low glycaemic index” as a message on the front-of-pack of legume-enriched bread could compel consumers to purchase these products instead of other less healthy bread options, such as white bread.

## 6. Nutrition Education and Practical Advice in the Kitchen

Skills-based nutritional education which aims to increase nutritional information to spread knowledge on pulses’ preparation in order to reduce their anti-nutritional factors, optimize cooking times, and create nutritionally balanced but more palatable dishes, represents an effective method to promote this food group. A cross-sectional study carried out on the Australian population [133] explored consumers’ perceived barriers and benefits of plant foods (including vegetables, cereals, pulses, nuts, and seeds), highlighting that participants considered pulses’ introduction into their diet a difficult task due to their taste, perceived time-consuming preparation required, and lack of knowledge on cooking recipes in order to make them palatable.

In support of the importance of being able to prepare meals independently, reducing the consumption of ready-made and processed foods, strong correlations are reported between healthy food preparation skills and better diet quality [134], as well as between time spent cooking and mortality [135]. Professionals can impart knowledge, tools, and attitudes to help patients acquire skills that will facilitate them to adopt a behavioral change. A 2015 review reported practical tools aimed at professionals and patients to facilitate the use of legumes in cooking and improve eating habits [27].

### 6.1. Culinary Approaches to Adopt towards the Use of Legumes

The suggestion for those who are not in the habit of using these kinds of foods in their diet is to start with lentils, because they are the easiest legume to cook: hulled lentils do not require soaking and have a short cooking time (20 to 30 min depending on the quality). Boiled lentils will keep in the refrigerator for 4–5 days, so a suggestion might be to overcook them and store them for use in various dishes in the following days.

#### 6.1.1. Pre-Treatment Prior to Cooking

Chickpeas and beans are legumes that require long soaking times (up to 24 h) because they have a seed coating matrix that makes it more difficult for water to permeate, so more time is needed to allow the water to soften the tegument and facilitate the reduction of phytates, tannins, polyphenols and alpha-galactosides. Two types of soaking can be carried out: the traditional soaking, which is longer (12–24 h), using large containers and using five parts of water for every part of the legume, and the quick soaking, which involves rinsing and boiling the seeds in water for two-to-three minutes. The fire is turned off and the beans are left to stand for three-to-four hours.

It is best to avoid adding bicarbonate, because although it makes the legumes softer, it tends to impoverish them, degrading the B vitamins and altering the taste. Slight acidification of the water, on the other hand, facilitates the germination and activation of phytases and, thus, the degradation of phytic acid: it is sufficient to add a tablespoon of lemon juice or apple cider vinegar per kg of legumes to obtain a good reduction in phytates. Excessive acidification, on the other hand, leads to hardening of the skin. After throwing away the soaking water and rinsing the pulses well, they are boiled, which can take up to more than two hours depending on the type of pulses; cooking is a long process but requires no special skills. The soaking time for the same type of legume may vary depending on the hardness of the seed (generally, older seeds are harder and require more soaking time). If a pressure cooker is used, the boiling time should be reduced by about a half. All pulses once boiled can be stored in the refrigerator for 3–4 days and used later to prepare numerous versatile recipes. They can also be bought already cooked: they are commercially available in tins. It is best to choose low-sodium products or rinse them well with running water before use.

#### 6.1.2. Legume Salads

A basic legume salad can be prepared with a combination of any type of cooked legume, together with chopped herbs, lemon juice or vinegar, olive oil, salt and pepper, chopped vegetables or fruit, cheese cubes or even slices of chicken breast or other meat, and dried fruit can be added. These salads can also be prepared a day in advance.

#### 6.1.3. Soups and Stews

Already-cooked legumes can be added to any kind of soup prepared as usual, or they can be boiled together with the vegetables and herbs and served either as they are or after blending them with an immersion blender or mixer to make creamy or velvety soups. Abdominal discomfort caused by the tegument fibers can be alleviated by using a vegetable mill to remove the peels.

#### 6.1.4. Single Dish

All legumes, when served together with a cereal, constitute a meal containing all the essential amino acids and are recommended as a protein choice, especially for vegetarians or vegans. Examples of such dishes are included in the Mediterranean diet: pasta and beans, rice and peas, pasta and lentils, and pasta with chickpeas. Or alternatively, by prolonging the cooking time of the lentils, obtaining a softer texture, they can be used to prepare purees or sauces with the addition of spices (curry, paprika, cumin, turmeric) and served together with a cereal such as brown rice, hulled millet, barley, or spelt; adding vegetables also improves the nutritional value of the dish and enhances the taste.

#### 6.1.5. Hummus

Hummus is a nutrient-dense sauce made from boiled and mashed chickpeas mixed with tahini (a sauce made from toasted sesame seeds), olive oil, lemon juice, and spices. With chickpeas as the main ingredient, it is a way to help the consumer maximize the nutrient-to-calorie ratio when choosing a sauce, due to its unique blend of health-beneficial ingredients (protein, fiber, vitamins, minerals), reducing the intake of saturated fats and refined carbohydrates when compared to other sauces and dressings, and also improving the appearance and taste of the dish [136].

#### 6.1.6. Utilization of Legume Flours

Legume flours can be used in the domestic preparation of vegetable burgers, meatballs, and croquettes and for the preparation of homemade pasta and for baked goods (bread, cakes, biscuits, taralli, breadsticks, crackers), mixed appropriately with other types of flours, to reduce technological drawbacks (especially for bread and cakes). Enrichment of these commonly consumed snacks, breakfast cereals and puffs, and baking and pasta food products with legumes represents a strategic approach to deliver essential nutrients and bioactive compounds to consumers [118,137,138]. However, it is important to ensure that optimum cooking/processing conditions are used during the development of legumes and their enriched products to ensure that the final legume-enriched products have appealing organoleptic properties, acceptable cooking qualities, and are nutritionally dense and with a high content of bioactive compounds [138,139]. Consequently, processing conditions involve temperatures spanning from 90 to 180 °C [113,140,141] and duration of 30 to 120 s. Ref. [142] has reported on the extrusion cooking of legume-enriched cereal snacks. In the case of soybean-enriched pasta, produced through cold extrusion, drying parameters involving 50 °C for 14 h [137,143] have been reported as adequate to deliver pasta with appreciable cooking losses and increased nutrient density.

Legumes cooking properly not only enhances their flavor and texture but also improves their digestibility and nutrient availability. Various factors such as soaking time, water pH, and cooking duration can influence the cooking outcomes of legumes. For instance, pre-treatment methods such as soaking can help soften the tegument and facilitate the reduction of phytates, tannins, polyphenols, and alpha-galactosides, which can improve the overall nutritional profile and digestibility of legumes [144]. Additionally, understanding the optimal cooking times for different types of legumes is essential to achieve the desired texture and doneness. The use of pressure cookers can significantly reduce cooking times [145], making legume preparation more convenient [2]. By following the correct cooking methods, legumes can be transformed into a variety of delicious and nutritious dishes, expanding their culinary versatility and promoting their inclusion in a healthy diet.

#### 6.1.7. Legume Sprouts

Whole seeds are used, soaked for 10–24 h, then spread in a layer on a large plate, tray or colander, covered with a damp kitchen cloth and left in a warm, dark place to sprout for 4–5 days, taking care to keep them moist, wetting the cloth daily. The sprouts can be kept in the refrigerator for 3–4 days and eaten raw, such as by adding to salads.

### 6.2. Effects of Culinary Approaches on the Nutritional Composition and Secondary Metabolites of Legumes

Cooking methods play a crucial role in determining the nutrient composition and bioavailability of legumes [31,139]. Various factors, such as cooking time, temperature, and pre-treatment methods, can affect the nutritional profile of legumes [138]. Here, we discuss the effects of different culinary approaches on the nutritional content and secondary metabolites of legumes.

Boiling and steaming: the boiling of legumes including chickpea, kidney bean, lentil, pea, and sword bean is associated with increased losses in vitamins E and K content [146]. In terms of macronutrient and minerals, reductions in carbohydrate, proteins, lipids, potassium, calcium, and iron were recorded for boiled and pressure-cooked pigeon peas with the stock decanted and undecanted [147]. Even though boiling improves the digestibility of legume proteins and destroys legume anti-nutrients, it is important to pay attention to the boiling duration to ensure the increased retention of essential nutrients. This can be practically carried out through an optimization study, with varied boiling temperatures and times and with participants recruited to conduct a sensory evaluation on boiled legumes. This could be complemented by the evaluation of the retained nutrients and the anti-nutrient composition. Cooking of legumes including green peas, yellow peas, chickpeas, and lentils using atmospheric and pressure boiling resulted in reductions in the concentration of total phenolic content and DPPH free radical scavenging activity (DPPH), when compared to their raw unprocessed form [148]. Considering legume pressure steaming, the steaming process resulted in increased retention of total phenolic content (TPC) and DPPH compared to boiling.

Soaking and boiling: soaking legumes can be used as a pre-treatment method before boiling legumes. Huma et al. [149] investigated the effects of cooking on nutritional and anti-nutritional factors of five different legumes including white kidney beans, red kidney beans, lentils, chickpeas, and white grams. In that study, the authors soaked the legumes in ordinary water, 2 percent sodium chloride solution, acetic acid, and sodium bicarbonate before cooking. The authors reported that the soaking and boiling of the legumes resulted in decreased nutritional values and antinutritional factors of the legumes. In fact, significant reductions in phytic acid and tannins were recorded for the legumes boiled after soaking in sodium bicarbonate solution [149]. Soaking legumes before cooking has been shown to reduce cooking time, to improve protein digestibility and to enhance the bioavailability of nutrients.

Germination/sprouting: germination/sprouting legumes further enhances their nutritional profile by increasing vitamins, minerals, and enzymes levels, while reducing anti-nutritional factors content [150]. Recently, Atudorei et al. [151] investigated the effects of legume germination in terms of nutritional and microstructural parameters, after two and four days, respectively. This experimental procedure included beans (*Phaseolus vulgaris*), lentils (*Lens culinaris Merr*.), soybeans (*Glycine max* L.), chickpeas (*Cicer aretinium* L.), and lupines (*Lupinus albus*). The authors reported that legumes sprouting resulted in increased protein and ash concentrations in beans, lentils, and chickpeas. As expected, a decrease in carbohydrate concentration was recorded for all the germinated legumes whereas fat decrease was recorded only in lentils, chickpeas, and lupines [151]. It must be emphasized that even though germination impacts positively on legumes, controlled germination must be carried out in order to avoid what has been reported on cereals, for example, in which uncontrolled sprouting results in a flour’s poor baking properties due to the over-production of hydrolytic enzymes [152].

Fermentation: spontaneous fermentation is a cost-effective method commonly used in several developing countries, including in Sub-Saharan Africa, to improve the nutritional density and digestibility of food materials. As for legumes, fermentation results in increased mineral bioavailability and the production of important bioactive compounds that have health-promoting benefits [153]. Additional benefits associated with legume fermentation include an increased production of essential compounds such as probiotics, organic acids, and antioxidants, which contribute to improving gut health and overall well-being [150,154]. Fermenting legumes leads to breaking down complex carbohydrates and proteins, making them easier to digest [150]. In terms of fermentation effects on secondary metabolites in legumes, it resulted in reductions of tannins, phytic acid, raffinose, alpha-galactoside e concentrations, and increased the production of legumes’ gamma amino butyric acid [154]. Fermentation can also be carried out by bacteria-mediated fermentation using species including *Pediococcus pentosaceus* SDL1409, *B. amyloliquefaciens* and LAB. The bacteria-mediated process results in increased production of flavanols, isoflavone aglycones, gallic acid, and increased antioxidant activity [154]. Fermentation effects on legumes include the stimulation of several endogenous enzymes activities including phytase, α-amylase, pullulanase, and other glucosidases, resulting in the breakdown of anti-nutritional factors and macronutrients into simple, digestible, and absorbable form by the body [153].

Taking into consideration the effects of different culinary approaches on the nutritional content of legumes and secondary metabolites, individuals can make informed choices about their cooking methods to maximize the health-promoting potential of legumes in their diet. Choosing the right culinary approach for legumes is crucial for maximizing their health-promoting effects. These approaches can impact the nutritional composition, bioavailability of nutrients, and overall health benefits derived from legume consumption. Optimal cooking methods preserve the nutritional content, while processing techniques such as soaking, sprouting, and fermentation enhance nutrient quality. By pairing legumes with complementary ingredients and considering the specific nutritional composition of the legumes’ variety, it is possible to further enhance their health benefits. However, it is important to note that culinary approaches are just one factor among many, and overall dietary patterns and individual health status also influence the health-promoting effects of legumes.

Table 5 shows factors to take into consideration when selecting culinary approaches for legumes, based on their nutritional content. These considerations can help optimize nutrient retention and enhance the overall nutritional quality of legume-based dishes. Factors such as heat exposure and cooking time, water usage, processing techniques, complementary ingredients, and the specific nutritional composition of the legumes used should be taken into account. Referring to nutritional databases or dietary guidelines can help to select an appropriate cooking method.

This table outlines essential factors to take into consideration when selecting culinary approaches for legume-based dishes based on their nutritional content. The recommendations highlight the importance of some aspects such as heat exposure and cooking time, water usage, processing techniques, complementary ingredients, and the specific nutritional composition of the legume variety being used. These factors can help optimize nutrient retention and enhance the overall nutritional quality of legume-based dishes.

### 6.3. Relationship between the Use of Right Culinary Approaches and Health-Promoting Effects of Legumes

The choice of right culinary approaches for legume processing helps to retain their nutritional content but also contributes to their health-promoting effects. The cooking methods can influence nutrients’ bioavailability and digestibility, as well as bioactive compounds release in legumes [153]. The health-promoting effects of a food are largely dependent on the bioavailability and bioaccessibility of the food’s essential nutrients and bioactive compounds [155]. Consequently, the adoption of appropriate culinary approaches promoting increased retention of essential nutrients and bioactive compounds, such as polyphenolic compounds, while reducing anti-nutritional factors can result in enhanced physiological effects and demonstrate health benefits. Bioactive compounds including phenolic acids, tannins, stilbenes, lignin, and coumarins, for example, have been reported to have the ability to inhibit the activity of the enzyme α-amylase which is involved in carbohydrate digestion [156]. Therefore, consuming legumes with increased polyphenol content may demonstrate a healthy effect in improving glycaemic response. It is therefore important to appreciate the nature and stability of the predominant nutrients and bioactive compounds of legumes, which could suggest the right culinary approach. For example, some bioactive compounds, such as flavonoids and phenolic compounds, are heat-sensitive and may be degraded during cooking. However, certain cooking methods, such as steaming or stir frying, can help preserve these bioactive compounds to a greater extent, leading to increased health benefits. Furthermore, combining legumes with other ingredients, such as herbs, spices, or vegetables, can synergistically enhance the overall antioxidant and anti-inflammatory properties of the dish. Culinary strategies regarding the use of water for fat-soluble vitamin-rich legume cooking and the use of cooking oil for legumes rich in water soluble vitamins would also need to be considered. This would lead to increased retention of the nutrients with less leaching into the cooking stock.

It is important to note that the health effects of culinary approaches may vary depending on the individual’s specific dietary needs, health status, and overall dietary patterns. Consulting with a healthcare professional or registered dietitian can provide personalized guidance on choosing the right culinary approaches for legume consumption to support individual health goals.

## 7. Conclusions

Legumes are recognized as nutritionally dense food raw materials that contain essential nutrients and bioactive compounds, which contribute to their significant health-promoting effects. Emerging issues including food neophobic tendencies or taboos, unclear dietary guidelines on legume consumption, as well as health concerns and socio-economic reasons including long cooking procedures, reduce legume consumption frequency. Pre-treatment methods including soaking, sprouting, and pulse electric field technology are effective in reducing the alpha-oligosaccharides and other anti-nutritional factors concentration, eventually lowering legume cooking time. The extrusion technology used for the development of innovative legume-enriched products including snacks, breakfast cereals and puffs, baked goods, and pasta represents a strategic way to promote legume consumption. Lastly, culinary skills such as legume salads, legume sprouts, stews, soups, hummus, and legume flour home-made cake recipe development could represent effective ways to promote legume consumption. Future research should consider investigating more innovative approaches to improve legume digestibility and reduce cooking time for more underutilized legumes. Additionally, gender-specific and age-segregated factors associated with legume consumption should be conducted. This could be essential to provide tailored nutritional interventions in these groups of consumers.

## 8. Recommendations

Firstly, it is crucial to address emerging issues such as food neophobic tendencies or taboos, unclear dietary guidelines on legume consumption, and health concerns. Policy makers should strive to create awareness and education campaigns to dispel misconceptions and promote the nutritional benefits of legumes. Clear and evidence-based dietary guidelines specific to legume consumption can guide individuals and healthcare professionals in making informed decisions.

Secondly, interventions focusing on pre-treatment methods, such as soaking, sprouting, and pulse electric field technology, can significantly reduce cooking time and enhance legume digestibility. Policies and practices should support the adoption of these techniques both at household and industrial levels. Moreover, research and development efforts should be encouraged to make these pre-treatment methods more accessible and feasible for widespread implementation.

Furthermore, the extrusion technology used in the development of innovative legume-enriched products represents an opportunity for policy and practice interventions. Policy makers should support research and funding initiatives aimed at promoting the production and availability of these products in the market. Additionally, collaborations with food manufacturers can help incentivize the development of healthier legume-based snack options and other convenient food choices.

Culinary skills and recipe development play a prominent role in promoting legume consumption. Policies and programs developed to emphasize culinary education and training, especially in schools and community settings, can empower individuals to incorporate legumes into their diets through the preparation of delicious and nutritious meals. Support for home cooking initiatives and the dissemination of legume-focused recipes can further encourage a wider adoption of legume-rich diets.

Finally, future research should promote the investigation of innovative approaches to improve legume digestibility and reduce cooking time for underutilized legumes. Policy makers should allocate funding for research projects in this area and collaborate with research institutions and agricultural organizations to explore strategies in order to enhance the market potential and nutritional value of these legume varieties.

Overall, a comprehensive policy framework should be established to address the challenges and support the opportunities for promoting legume consumption. By implementing evidence-based policies, supporting research and development, and promoting culinary education, policy makers and practitioners should work synergically to promote the inclusion of legumes into diets and, hence, improve public health outcomes.

## Figures and Tables

**Figure 1 foods-12-02265-f001:**
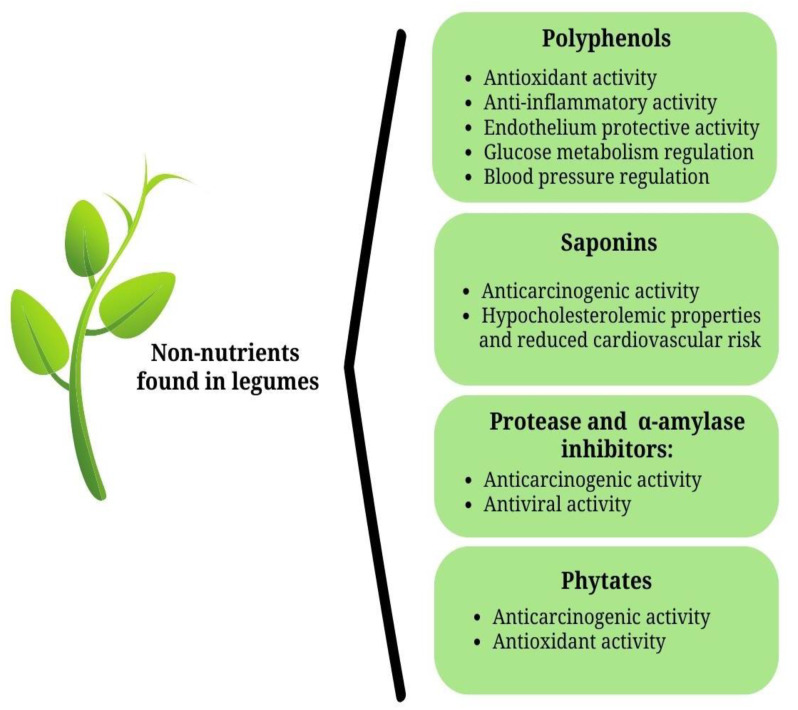
Biological activity of the minor components of legumes. The figure shows the biological activity of the minor components of legumes and their potential health effects.

**Figure 2 foods-12-02265-f002:**
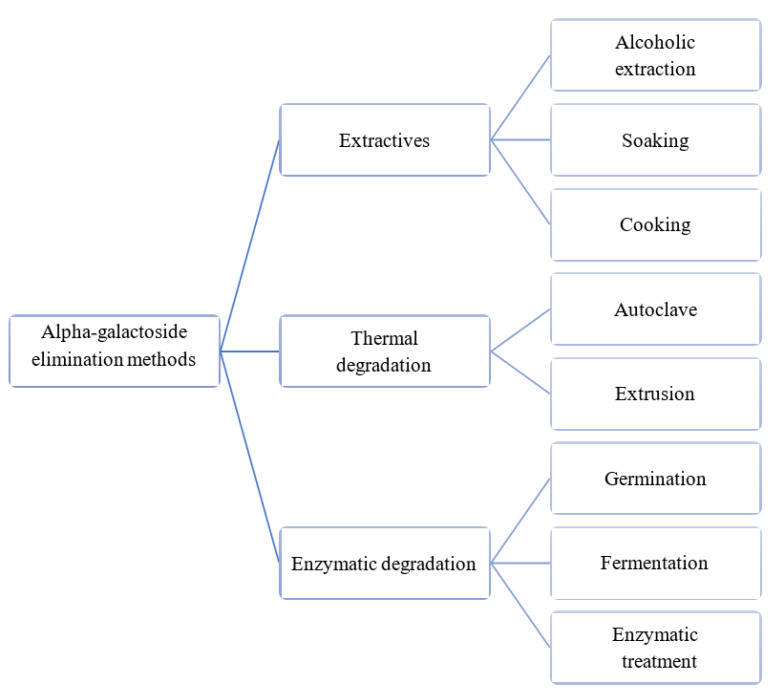
Methods to reduce the cooking time of legumes. The figure presents different techniques or methods for shortening the cooking time of legumes.

**Figure 3 foods-12-02265-f003:**
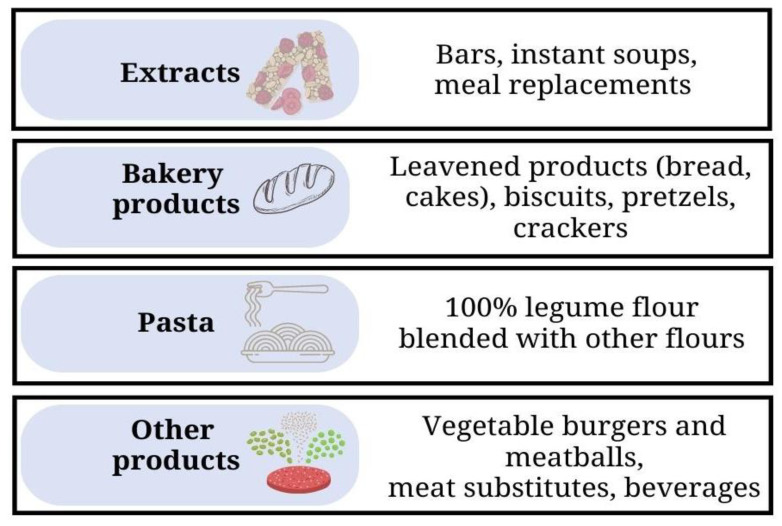
Main processed products that can be obtained from legume meal. The figure shows the various processed products that can be produced from legume meal.

**Table 1 foods-12-02265-t001:** Nutrient composition of dried raw legumes (per 100 g) [28].

Legumes	Water (%)	Energy (kcal)	Protein (%)	Carbohydrates (%), Excluding Fiber)	Total Fibers (%)	Soluble Fibers (%)	Insoluble Fibers (%)	Lipids (%)	Resistant Starch (%)	Oligosaccharides (%)	Other Non-Digestible Fibers (%)
Common beans	11.9	333	23.6	45.0	15.0	2.0	13.0	1.2	4.8	1.9	0.8
Lentils	11.8	353	24.6	52.2	10.8	1.0	9.8	1.1	2.9	1.8	0.8
Chickpeas	10.7	364	19.0	44.0	17.0	3.4	13.6	6.0	1.7	2.7	1.3
Peas	10.7	338	25.0	44.0	16.0	2.0	14.0	1.2	0.8	1.6	0.7
Broad beans	11.3	341	26.0	33.0	25.0	1.8	23.2	1.5	1.8	1.0	2.2
Soybeans	8.5	446	36.0	21.0	9.0	2.0	7.0	20.0	1.4	4.9	0.6

This table displays the nutrient composition of dried raw legumes per 100 g for six different types of legumes.

**Table 2 foods-12-02265-t002:** Concentration of alpha-galactosides and raffinose (% dry matter) in 100 g of raw legumes.

Legume	Stachyose	Verbascose	Raffinose	Ajugose
Common beans	3.3	2.6	0.4	0.2
Lentils	0.2	0.2	0.0	0.0
Chickpeas	1.0	0.8	0.2	0.0
Peas	0.4	0.2	0.2	0.0
Broad beans	0.2	0.3	0.0	0.0
Soybeans	1.5	0.6	0.3	0.0

This table shows the % dry matter of different types of alpha-galactosides (stachyose, verbascose, and ajugose) and the trisaccharide raffinose in 100 g of raw legumes. Alpha-galactosides and raffinose are types of oligosaccharides found in legumes that can cause intestinal discomfort in some individuals. The data in this table were sourced from Mudryj et al. [43].

**Table 3 foods-12-02265-t003:** Main possible toxic effects of anti-nutritional factors in legumes [87].

Anti-Nutritional Factor	Legume(s) That Contain It	Possible Health Effects
Phytic acid	Soybeans, chickpeas, lentils, kidney beans, black beans	Diarrhea, nausea, vomiting, abdominal pain, impaired nutrient absorption
Lectins	Kidney beans, lima beans, peanuts	Reduced protein digestion, decreased protein utilization
Protease inhibitors	Soybeans, kidney beans, lima beans, peanuts	Impaired mineral absorption, reduced bioavailability of dietary minerals
Saponins	Chickpeas, lentils, peas	Reduced protein digestion, decreased protein utilization, impaired nutrient absorption
Tannins	Kidney Beans, lima beans, mung beans	Hemolysis, intestinal irritation, decreased nutrient absorption
Lathyrogens	Chickpeas, lentils, peas	Flatulence, abdominal bloating, decreased nutrient absorption
Oligosaccharides	Chickpeas, kidney beans, lentils, navy beans	Reduced protein digestion, decreased protein utilization
Cyanogens	Lima beans, fava beans	Neurotoxicity, paralysis
Phytoestrogens	Soybeans	Hemolytic anemia, favism
Trypsin Inhibitors	Soybeans, lima beans, kidney beans, peanuts	Autoimmune response, impaired nutrient absorption

The table shows anti-nutritional factors commonly found in legumes and their possible health effects.

**Table 4 foods-12-02265-t004:** Nutrient content of dry legumes.

Nutrient	100 g Dry Legumes	RDA (Male)	RDA (Female)	AI
Protein (g/day)	20–30	56	46	-
Fiber (g/day)	8–16	38	25	-
Folate (μg/day)	300–600	400	400	320–400
Iron (mg/day)	2.5–7	8	18 (premenopausal)	8 (postmenopausal)
Magnesium (mg/day)	70–130	420	320	310–420
Phosphorus (mg/day)	250–500	700	700	-
Potassium (mg/day)	500–1000	3400	2600	2000–3100

**Table 5 foods-12-02265-t005:** Factors to consider for choosing culinary approaches based on nutritional content.

Factors	Recommendations
Heat exposure and cooking time	Opt for methods with shorter cooking times or lower heat to preserve nutrients. Examples include steaming, stir-frying, or sautéing legumes. Avoid overcooking or prolonged high-heat cooking methods which may lead to nutrient losses.
Water usage	Use methods that involve limited water contact and volumes to minimize nutrient leaching. Consider using boiling methods such as pressure cooking, where less water is required. Additionally, using the soaking water (for legumes that require soaking) in boiling procedures can help retain some water-soluble nutrients.
Processing techniques	Soaking, sprouting, and fermentation can enhance nutrient quality. Soaking legumes before cooking reduces cooking time and improves digestibility. Sprouting legumes increases nutrient availability and reduces anti-nutrients content. Fermenting legumes enhances their nutritional profile and promotes the growth of beneficial bacteria.
Complementary ingredients	Pair legumes with foods rich in vitamin C or fat sources to enhance nutrients absorption. Vitamin C facilitates the legumes’ non-heme iron absorption, while pairing legumes’ consumption with a source of fat can improve the absorption of fat-soluble vitamins. Including citrus fruits, bell peppers, or tomatoes in legume dishes can provide vitamin C, while adding olive oil, avocado, or nuts can contribute to the fat content.
Consider nutritional composition of variety	Take into account the specific nutritional composition of the variety of legume being used. Different legumes may vary in their nutrient profiles, so understanding their individual characteristics can guide the selection of appropriate cooking methods. Referring to nutritional databases or dietary guidelines can provide valuable informations on various legume varieties’ nutrient contents.

## Data Availability

Not applicable.

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
