# Peer review of "Sustainable Strategies for Increasing Legume Consumption: Culinary and Educational Approaches"

_foods, 2023, doi:10.3390/foods12112265_

Round 1

Reviewer 1 Report

The manuscript “Sustainable Strategies for Increasing Legume Consumption: Culinary and Educational Approaches”. This review aims to highlight the nutritional and health effects associated with legume consumption, strategies to improve their digestibility and nutritional profile. However, the manuscript still need more improvement.

1. What are the effects of different culinary approaches on the nutritional content of legumes and secondary metabolites of plants?

2. How to choose the right culinary approaches according to the nutritional content?

3. What is the relationship between right culinary approaches and health-promoting effects?

4 Taking different products rich in legumes (e.g. snacks, breakfast cereals and puffs, baking and pasta) as an example, it would be more instructive to clearly indicate the correct cooking methods and conditions, as well as possible theoretical results.

None

Author Response

The manuscript “Sustainable Strategies for Increasing Legume Consumption: Culinary and Educational Approaches”. This review aims to highlight the nutritional and health effects associated with legume consumption, strategies to improve their digestibility and nutritional profile. However, the manuscript still need more improvement.

  1. Comment: Pages 25-26 What are the effects of different culinary approaches on the nutritional content of legumes and secondary metabolites of plants?

Response to comments: We appreciate the reviewer's insightful comment regarding the effects of different culinary approaches on the nutritional content of legumes and secondary metabolites of plants. Indeed, culinary techniques can significantly impact the nutrient composition and bioactive compounds in legumes, thereby influencing their potential health benefits.

To address this aspect, we have included a new section in the manuscript “6.2 Effects of Culinary Approaches on the Nutritional Composition and Secondary metabolites of Legumes”. This section is specifically dedicated to discussing the effects of culinary approaches on the nutritional content of legumes and secondary metabolites of plants.  We believe that the inclusion of this new section will provide a more comprehensive understanding of how culinary techniques can influence the nutritional value and bioactive compounds in legumes. It emphasizes the importance of adopting appropriate cooking methods to retain or enhance the nutritional content and secondary metabolites of legumes, thereby maximizing their potential health benefits.

Thank you for the valuable suggestion, and we have incorporated it into the revised manuscript accordingly.

  1. Comment: How to choose the right culinary approaches according to the nutritional content?

Response to comment: We thank the reviewer for raising the question regarding the relationship between the right culinary approaches and health-promoting effects of legumes. In summary, the choice of culinary approaches significantly impacts the health benefits of legumes. Proper cooking methods help retain essential nutrients, while processing techniques like soaking, sprouting, and fermentation enhance nutritional quality. Complementary ingredients can further enhance nutrient absorption. Although other factors also influence health outcomes, adopting the right culinary approaches optimizes the nutritional content and bioavailability of legumes, maximizing their potential health benefits. 

We added a table to the manuscript. “Table 5. Factors to Consider for Choosing Culinary Approaches Based on Nutritional Content”

  1. Comment: Page 27-28 What is the relationship between right culinary approaches and health-promoting effects?

Response to comment: We appreciate the reviewer's comment regarding the relationship between the right culinary approaches and health-promoting effects of legumes. Indeed, the culinary approaches employed can significantly influence the potential health benefits associated with legume consumption. We have added a new section to the manuscript “6.3 Relationship between the use of right culinary approaches and health-promoting effects of legumes

  1. Comment:  Page 24 Taking different products rich in legumes (e.g. snacks, breakfast cereals and puffs, baking and pasta) as an example, it would be more instructive to clearly indicate the correct cooking methods and conditions, as well as possible theoretical results.

Response to comment: We thank the reviewer for their valuable suggestion regarding the inclusion of clear instructions on correct cooking methods and conditions for various legume-rich products, such as snacks, breakfast cereals, puffs, baking, and pasta. We agree that providing explicit guidance in this regard would enhance the instructiveness of our manuscript and allow readers to maximize the health-promoting effects of legumes in their culinary endeavors. In response to the reviewer's comment, we have incorporated new sections (6.2 , 6.3 and Recommendations) about the importance of understanding the correct cooking methods and conditions for legumes to maximize their health-promoting effects. 

We believe that these additions will address the reviewer's suggestion and further enhance the value and applicability of our manuscript. Thank you once again for your valuable input.

Reviewer 2 Report

My comments are as follows:

1. This review aims to highlight the nutritional and health effects associated with legume consumption, strategies to improve their digestibility and nutritionaprofile” should be placed at the end of the abstract to make it more logical.

2. Add "%" after each metric in the first row in Table 1.

3. Figure 1 and Figure 3 are not clear enough to be seen.

4. The first sentence in the conclusion completely coincides with the first sentence of the abstract. It is recommended to change the expression without changing the meaning.

5. Please adjust the paragraph spacing to be consistent.

My comments are as follows:

1. This review aims to highlight the nutritional and health effects associated with legume consumption, strategies to improve their digestibility and nutritionaprofile” should be placed at the end of the abstract to make it more logical.

2. Add "%" after each metric in the first row in Table 1.

3. Figure 1 and Figure 3 are not clear enough to be seen.

4. The first sentence in the conclusion completely coincides with the first sentence of the abstract. It is recommended to change the expression without changing the meaning.

5. Please adjust the paragraph spacing to be consistent.

Author Response

1.Comment: Page 1 “This review aims to highlight the nutritional and health effects associated with legume consumption, strategies to improve their digestibility and nutritional profile” should be placed at the end of the abstract to make it more logical.

Response to comment: We thank the reviewer for the valuable advice. We have added the sentence at the end of the abstract. The new sentence reads as “This review aims to highlight the nutritional and health effects associated with legume consumption, strategies to improve their digestibility and nutritional profile. Additionally, proper educational and culinary approaches finalised to improve legumes intake, is discussed.” 

  1. Comment: Page 3 Add "%" after each metric in the first row in Table 1.

Response to comment: The authors have fixed the first row of the table as requested by adding “%” to the column headings. 

  1. Comment: Figure 1 and Figure 3 are not clear enough to be seen.

Response to comment:  We have replaced the two images with two new figures

  1. Comment: Page 28 The first sentence in the conclusion completely coincides with the first sentence of the abstract. It is recommended to change the expression without changing the meaning.

Response to comment: We apologise for the repeated sentence. We have changed the first sentence of the conclusion as follows: “Legumes are recognised as nutritionally dense food raw materials that contain essential nutrients and bioactive compounds, which contribute to their significant health-promoting effects..”

  1. Comment: Please adjust the paragraph spacing to be consistent.

Response to comment: We tried to fix the formatting. 

Response to comment: We appreciate the reviewer's feedback and have made the necessary adjustments to the manuscript as requested. The paragraph spacing has been reviewed and adjusted to ensure consistency throughout the document. We believe that this modification enhances the overall readability and visual presentation of the paper. Thank you for bringing this to our attention, and we are grateful for your valuable input in improving the quality of our manuscript.

Reviewer 3 Report

On balance, this is an excellent review article. It covers a very wide range of topics and provides useful advice for researchers and nutritionists.

There is almost too much information, as the manuscript comes across as something of a travelogue. It could benefit from a more specific framework organized around major themes, with nested subthemes.

The global dimensions of the text are a major strength of the manuscript, with intriguing political and socio-cultural contexts.

A related point is that the manuscript would be strengthened by a more extensive conclusion that provides specific statements about implications for policy and practice.

The quality of exposition is mostly high, but the manuscript will benefit from a careful reading with attention to the level of presentation required for publication and to facilitate readers' comprehension.

Author Response

On balance, this is an excellent review article. It covers a very wide range of topics and provides useful advice for researchers and nutritionists. There is almost too much information, as the manuscript comes across as something of a travelogue. It could benefit from a more specific framework organized around major themes, with nested subthemes.

Thank you for this comment. The authors have revised the presentation of the information in a manner that is more focused and organized as suggested. For example, under the section “3. Barriers towards the consumption of legumes”, we have grouped the sub-sections “3.1.1 Food neophobic tendencies, 3.1.2 Food taboos and 3.1.3 Socio-economic factors” under the major Theme titled “3.1 Phychosocial and socio-economic reasons”. 

The global dimensions of the text are a major strength of the manuscript, with intriguing political and socio-cultural contexts.

We appreciate the reviewer's positive feedback regarding the global dimensions of our manuscript. 

A related point is that the manuscript would be strengthened by a more extensive conclusion that provides specific statements about implications for policy and practice.

We thank the reviewer for the valuable feedback. Based on the suggestion, we have revised the manuscript and included these additions as part of recommendations from this review in a new final section entitled “Recommendations”.

Comments on the Quality of English Language

The quality of exposition is mostly high, but the manuscript will benefit from a careful reading with attention to the level of presentation required for publication and to facilitate readers' comprehension.

We appreciate the reviewer's insightful comments. We have carefully reviewed the manuscript, taking into consideration the level of presentation required for publication and the need to facilitate readers' comprehension. Furthermore, a native English speaker has conducted a thorough review to ensure the clarity and accuracy of the text. Thank you for highlighting the importance of these aspects, and we have made the necessary revisions to improve the quality of exposition.

Round 2

Reviewer 1 Report

All the questions have been solved.

None

Reviewer 2 Report

Yes, the authors have revised the manuscript accordingly. Thank you.

Yes, the authors have revised the manuscript accordingly. Thank you.

Reviewer 3 Report

The extensive revisions have improved the manuscript considerably.

English usage is strong generally.

Final editorial checks will be required to ensure that revisions have been consistent with professional technical language.